# Genomic surveillance of COVID-19 cases in Beijing

Pengcheng Du[1,2,11], Nan Ding [1,2,11], Jiarui Li[1,2,11], Fujie Zhang [1,11], Qi Wang[1,11], Zhihai Chen [1,11], Chuan Song[1,2], Kai Han[1,2], Wen Xie[1], Jingyuan Liu[1], Linghang Wang[1], Lirong Wei[1], Shanfang Ma[1], Mingxi Hua[1,2], Fengting Yu[1], Lin Wang[3], Wei Wang[3], Kang An[4], Jianjun Chen [5,6], Haizhou Liu [6], Guiju Gao[1], Sa Wang[1], Yanyi Huang [7], Angela R. Wu [8], Jianbin Wang [9✉], Di Liu [5,6,10✉], Hui Zeng [1,2✉] & Chen Chen [1,2✉]

The spread of SARS-CoV-2 in Beijing before May, 2020 resulted from transmission following both domestic and global importation of cases. Here we present genomic surveillance data on 102 imported cases, which account for 17.2% of the total cases in Beijing. Our data suggest that all of the cases in Beijing can be broadly classified into one of three groups: Wuhan exposure, local transmission and overseas imports. We classify all sequenced genomes into seven clusters based on representative high-frequency single nucleotide polymorphisms (SNPs). Genomic comparisons reveal higher genomic diversity in the imported group compared to both the Wuhan exposure and local transmission groups, indicating continuous genomic evolution during global transmission. The imported group show region-specific SNPs, while the intra-host single nucleotide variations present as random features, and show no significant differences among groups. Epidemiological data suggest that detection of cases at immigration with mandatory quarantine may be an effective way to prevent recurring outbreaks triggered by imported cases. Notably, we also identify a set of novel indels. Our data imply that SARS-CoV-2 genomes may have high mutational tolerance.

[1] Beijing Ditan Hospital, Capital Medical University, Beijing 100015, People's Republic of China. [2] Beijing Key Laboratory of Emerging Infectious Diseases, Beijing 100015, People's Republic of China. [3] MGI, BGI-Shenzhen, Shenzhen 518083, People's Republic of China. [4] BGI-Genomics, BGI-Shenzhen, Shenzhen 518083, People's Republic of China. [5] CAS Key Laboratory of Special Pathogens, Wuhan Institute of Virology, Center for Biosafety Mega-Science, Chinese Academy of Sciences, Wuhan 430071, People's Republic of China. [6] National Virus Resource Center, Wuhan Institute of Virology, Chinese Academy of Sciences, Wuhan 430071, People's Republic of China. [7] Biomedical Pioneering Innovation Center (BIOPIC), Beijing Advanced Innovation Center for Genomics (ICG), College of Chemistry and Molecular Engineering, Peking-Tsinghua Center for Life Sciences, Peking University, Beijing, People's Republic of China. [8] Division of Life Science and Department of Chemical and Biological Engineering, Hong Kong University of Science and Technology, Hong Kong, SAR, People's Republic of China. [9] School of Life Sciences, Tsinghua-Peking Center for Life Sciences, Beijing Advanced Innovation Center for Structural Biology (ICSB), Chinese Institute for Brain Research (CIBR), Tsinghua University, Beijing 100084, People's Republic of China. [10] University of Chinese Academy of Sciences, Beijing 101409, People's Republic of China. [11] These authors contributed equally: Pengcheng Du, Nan Ding, Jiarui Li, Fujie Zhang, Qi Wang, Zhihai Chen. ✉email: jianbinwang@tsinghua.edu.cn; liud@im.ac.cn; zenghui@ccmu.edu.cn; chenchen1@ccmu.edu.cn

The coronavirus disease 2019 (COVID-19) is an emerging disease caused by severe acute respiratory syndrome coronavirus 2 (SARS-CoV-2). The main symptoms of this respiratory illness include cough, fever, shortness of breath, ranging from mild to severe[1–4]. Since cases were firstly reported in December 2019, the disease has caused over 34 million infections and over one million deaths in 235 countries and regions[5]. Based on the spatio-temporal distribution of new cases, two periods can be recognized from the pandemic development[6]. From the end of 2019 to February 2020, the outbreak was mainly domestic spreading within China, and the peak of new cases was recorded on February 12[7]. Upon implementation of a series of interventions, including routine screening and quarantine of travelers, self-isolation and detection, contact restrictions, and social distancing measures, the number of new infections in China has decreased, and the disease has been well controlled[8–10]. At the end of February, sporadic cases and outbreaks started to occur in Europe and Eastern Mediterranean. The outbreak developed into a new stage when the number of infections in Europe and America overtook China.

On March 11, the World Health Organization officially declared the outbreak of a global pandemic, as 114 countries reported over 118,000 infections. As Europe and the United States became the epicenters of disease outbreak, China started to experience case importations from abroad. The emphasis on disease control in China has then shifted from community-level interventions to control borders and immigration. The strict measurements of disease control in China then turned into the detection and quarantine of imported infections. Further local transmission from these imported cases was successfully minimized by the implementation of screening measures, including body temperature measurements and real-time polymerase chain reaction (RT-PCR) tests, on inbound travelers at ports of entry such as airports. As of May 30, China has recorded over 1700 imported cases, including Chinese travelers, Chinese residents overseas, and foreign travelers (http://www.nhc.gov.cn/yzygj/s7653p/202003/46c9294a7dfe4cef80dc7f5912eb1989.shtml).

In this study, we perform deep SARS-CoV-2 genome sequencing followed by bioinformatic analysis on the clinical samples we collected during the abovementioned two periods in Beijing. We summarize the characteristics of the viral genome sequences, including several notable changes in the viral genomic features during these periods.

## Results

### Domestic and global imported transmission of SARS-CoV-2.
We admitted 273 SARS-CoV-2 positive patients to our hospital from January 29 to April 17, 2020, accounting for 46.0% of the total 593 reported cases in Beijing during this time (Fig. 1a, b and Supplementary Fig. 1). The mean age of the cases in our hospital (137 males and 136 females) was $37.9 \pm 19.8$ years (QRI: 22.5–51.0). They were of varying disease symptoms, including 67 (24.5%) mild cases, 163 (59.7%) moderate cases, 43 (15.8%) severe or critical cases. Our cohort included 55 cases of returning travelers from Wuhan from January 20 to February 9 (termed Wuhan group). Also included were 81 patients who were infected by local transmission from the Wuhan group. A third-generation patient was also observed in a family outbreak on January 28. None of the family members had been to Wuhan, but one of them could be linked to another patient with Wuhan exposure. Eleven cases had no clear epidemiological relationships with other cases. From February 29 to April 17, a total of 121 overseas travelers (accounting for 69.5% of imported cases in Beijing) with positive nucleic acid test results were admitted to our hospital (imported group), including 101 from Europe and 15 from North

America (Fig. 1a). Due to strict screening and quarantine policies at the border, only four local cases (one confirmed on March 23, three on April 15) by close contact with imported cases emerged by April 17.

### Genetic clusters of SARS-CoV-2 in two transmission periods.
To uncover the genomic characteristics and diversities of the virus from imported cases in Beijing, we collected samples including pharyngeal swabs, sputum, and fecal samples from 178 patients. Applying a deep SARS-CoV-2 genome sequencing strategy[11,12], we obtained 34.6 (IQR: 14.1–94.6) million reads per sample overall, of which 14.1% (IQR: 1.1–58.8%) were viral reads. After removing the low-quality reads and removing the samples in which viral genome coverage is below 85%, we obtained near-complete viral genomes of 102 cases (20 Wuhan, 43 Wuhan-related local, 39 Imported, Fig. 1c) with a median depth of 8872× (Fig. 1d, Supplementary Table 1). The viral reads in the 102 corresponding samples comprised 40.2% (IQR: 12.2–80.0%) of the total reads obtained. Using these data, we obtained 102 high-quality genomes and identified 492 high-quality variations that occurred in 221 high-frequency single nucleotide polymorphism (SNP) sites (Supplementary Data 1).

We also performed genomic variation analysis on these data in conjunction with additional publicly available viral genomes (GISAID, Supplementary Data 2)[13]. By aligning the 4118 viral genome sequences from GISAID to the reference genome of Wuhan-Hu-1 (accession: NC_045512.2), we detected 2752 variations that occurred in 2669 SNP sites. Using the dynamic SARS-CoV-2 nomenclature system[14], we identified a broad representation of viral diversity in our data (Supplementary Fig. 3), likely because strong selection for any particular variation has not happened during the early stage of an epidemic. To simplify the lineage representation of our data, we tried to cluster the sequences using anchor mutations. Seventeen high-frequency SNPs (hfSNPs) that appeared >5% of the 4,220 viral genomes were identified from 2795 SNPs sites (removed repeated sites between 221 sites in our data and 2669 sites from GISAID data) as the anchor mutations (Supplementary Fig. 2). Of this 17 hfSNPs, 12 are non-synonymous mutations in different genes: five in ORF1ab, one in S, two in ORF3a, one in ORF8, and three in N. Of the remaining five hfSNPs, four are synonymous mutations in ORF1ab, and one locates in 5′UTR (Supplementary Fig. 2 and Supplementary Table 2). Using these hfSNPs, we clustered the SARS-CoV-2 sequences into seven groups (Fig. 2a and Supplementary Fig. 3). To better illustrate the relationships among seven clusters, we visualized the clusters using a minimum-spanning tree between each cluster (Fig. 2b). This structure is high concordance with the phylogenic tree with a standard nomenclature system, except that it is now more balanced to represent our data[14].

### Spatio-temporal analysis on seven genetic clusters.
Based on our analysis, during the early outbreak in Wuhan and the first period for Beijing infections, the dominant viruses are those in clusters C1 and C2; these were previously defined as L-type and S-type, respectively[15]. The phylogenetic analysis well presented the topology of seven clusters in Beijing, that C3 (Lineage A.1) was derived from C2 (Lineage A and A.2); C4 (Lineage B.2.1) was derived from C1 (Lineage B and B.2); and C5 (Lineage B.1) was derived from C1, followed by the emergence of C6 (Lineage B.1.1), and C7 (Lineage B.1) from C5 (Fig. 2c, and Supplementary Figs. 3, 4). To illustrate the geographical characteristics of the seven clusters, we analyzed the frequency of occurrence for each cluster along with the date of the global outbreak based on 4,220 genome data, including our primary data and publicly available

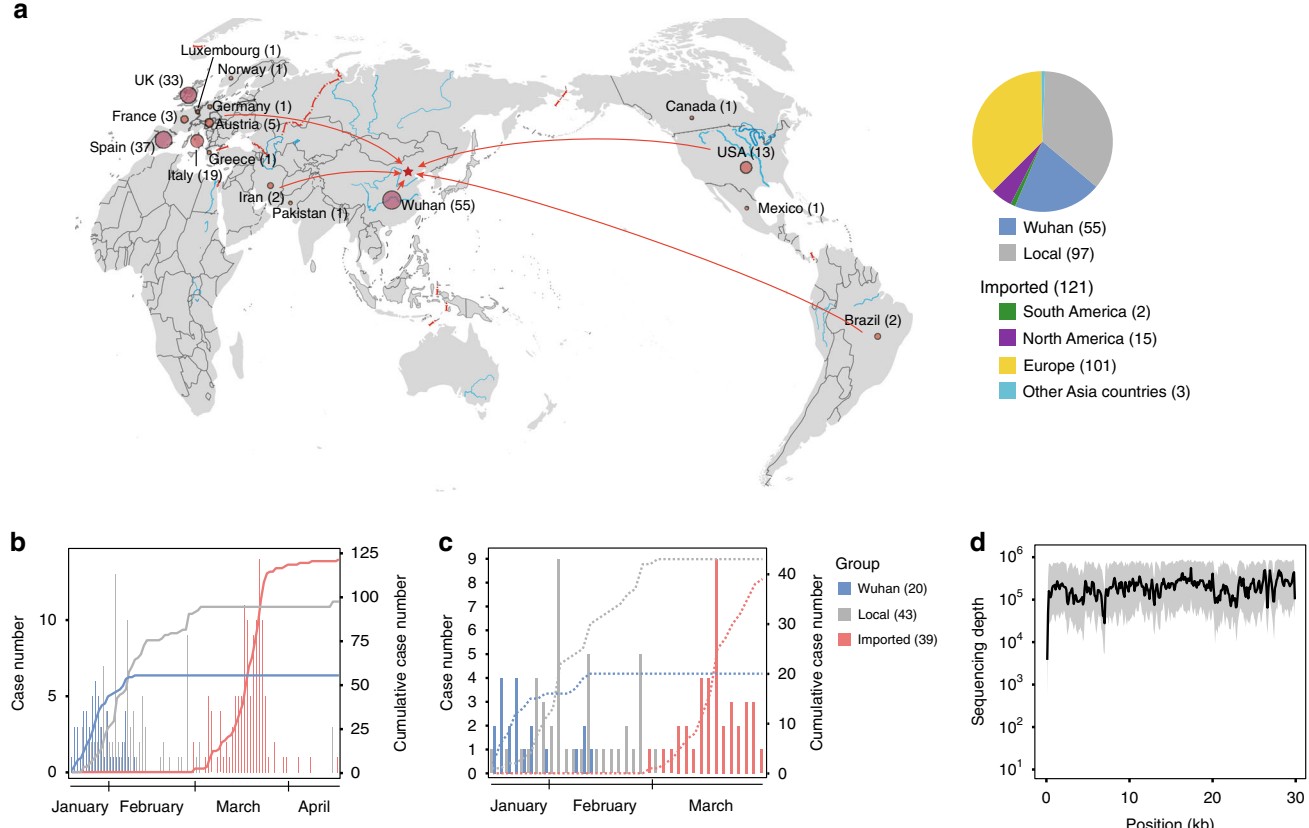

**Fig. 1 The COVID-19 cases in Beijing and sequenced in this study. a** The geographic sources of the cases in our hospital. **b** The number of cases in Wuhan group (from or passed by Wuhan), local group (onset locally and without Wuhan or overseas travel history), and imported group (overseas imported) in our hospital. The histograms represent the daily case numbers and the lines show cumulative case numbers in the three groups. **c** The number of well-sequenced cases (n = 102). The histograms represent the sequenced case numbers on each day and the orange line shows the cumulative case numbers. **d** The average sequencing coverage among the sequenced cases (n = 102). The shadow represents the lower and upper quartiles. Source data are provided as a Source Data file.

data (Supplementary Fig. 5), and found that the time period of peak occurrence frequency for each cluster is consistent with the reported infection numbers. C1 and C2 were found as the majority (~50%) of virus strains from Asia; Cluster C3 and C7 comprised more than half of the viruses from North America; and Clusters C4, C5, and C6 were dominantly (>70%) from Europe (Supplementary Fig. 6). For the clusters that dominate the domestic infection cases, the C2 cluster first peaked on January 30, then again on March 9, and for C1, two major peaks were observed on February 17 and March 11. For the clusters that are more frequently found in the imported infection cases, cluster C3–C7 peaked from March 13 to March 18, which is associated with the timing of outbreaks in Europe and North America. Notably, most countries, including the UK, the US, and China, contained more than two clusters of virus (Supplementary Fig. 6), indicating a complex genetic distribution during the COVID-19 pandemic.

By mapping these seven clusters to the viral spread in Beijing, we clearly observed two periods of infections (community-linked and international-linked) from independent clusters. In the period of community-linked cases, clusters C1 and C2 were present (Fig. 2c, d). We categorized the patients into two groups based on their exposure history, one with Wuhan exposure history and the other without. More C2 types were observed in patients without Wuhan-exposure history. In contrast, in the period of dominance with international-linked cases, the cases consist of more diverse viral cluster types, implying higher genomic diversity outside China during global transmission.

note, clusters C1 and C2 can still be found in the international-linked cases in March 18 and March 16, respectively, suggesting that clusters C1 and C2 have not been completely displaced outside China.

**Inter- and intra-genomic variations in Beijing**. To further elucidate detailed genomic mutations, we analyzed the occurrence of SNPs and intra-host single nucleotide variations (iSNVs) of the virus genomes, and compared them between the Wuhan exposure group, local transmission group, and imported group (Fig. 3a). In total, we identified 492 variations in 221 SNP sites within the 102 genomes we sequenced, which is roughly 3–7 variations per patient; of these, 74 are synonymous, 138 are non-synonymous substitutions (Supplementary Data 1). There are 11 protein- and polypeptide-encoding genes that contain more than 10 SNPs per gene. We calculated the non-synonymous to synonymous SNPs ratio (dN/dS ratio) for these highly poly-morphic regions (Supplementary Table 3). Intriguingly, the S protein-coding gene has a dN/dS ratio of 2.2. None of the 221 SNPs showed up in more than 15% of the 20 Wuhan exposure group genomes, which implies that SNPs found in the Wuhan cohort during the end of 2019 to February 9, 2020, were randomly generated, and no genotypes exhibited any advantages in transmission (Fig. 3b). In contrast, during the local transmission in Beijing in the first period, two variations representing the A lineage (C8782T and T28144C) showed up at an elevated frequency (>30%). C8782T located in *ORF1ab* was synonymous,

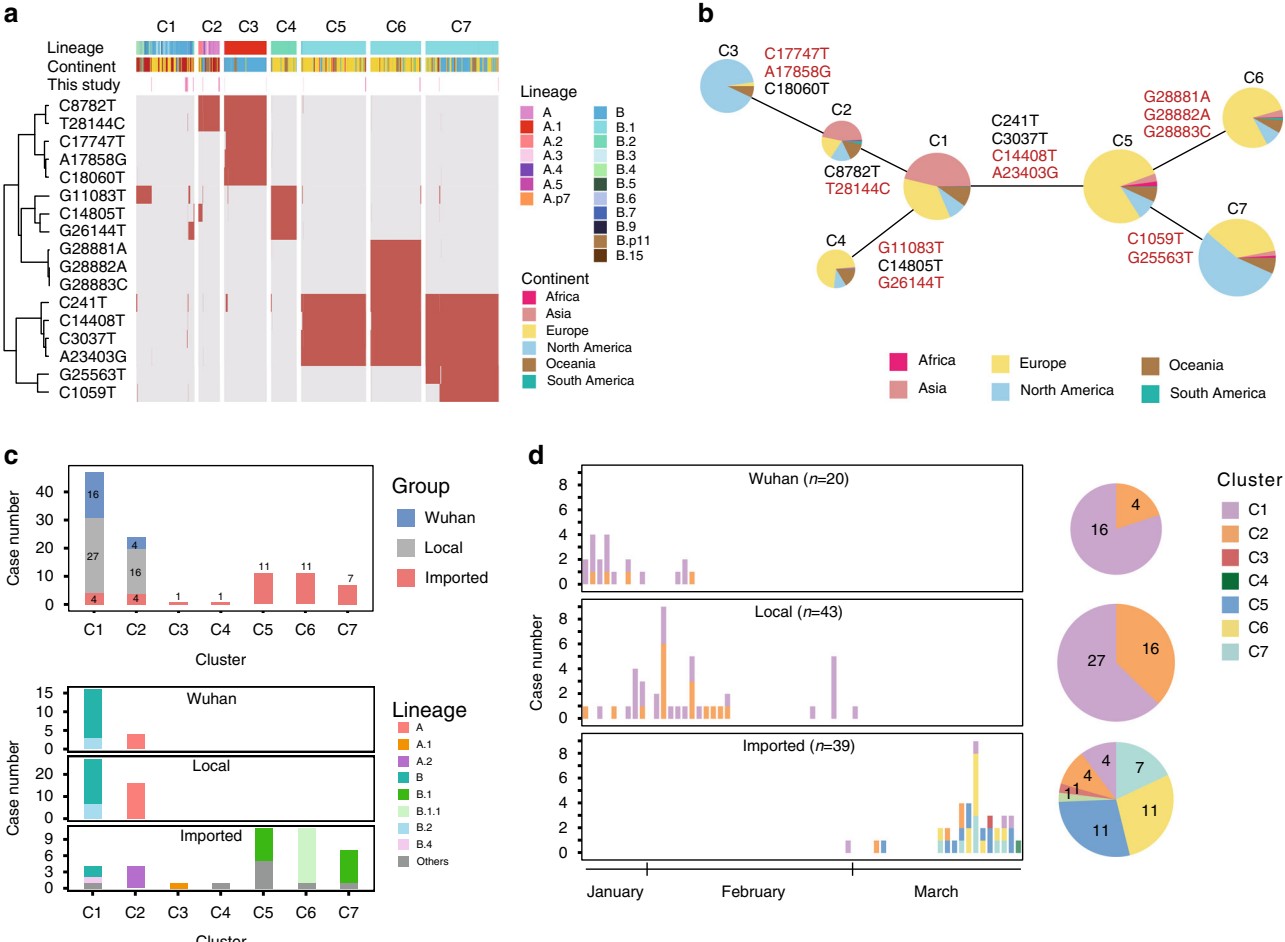

**Fig. 2 Genotype analysis and lineage determination of the viral genomes and the difference among the three groups. a** The viral genotypes from our data and public genome data. The heatmap shows genome clustering based on 17 high-frequency SNPs. The purple lines above the heatmap indicate the genomes sequenced in this study. The sublineages are marked on the top[16], and the geographic sources of the genomes are marked below the lineage. **b** The minimum-spanning tree of the seven clusters. The genomes from different continents are displayed on the pie by colors. The SNP differences are displayed on the lines between two clusters, and those caused by non-synonymous mutations are in red. **c** The histogram on the top displays the numbers of genomes we sequenced from each group in each cluster, and that at the bottom displays the early major lineages[16] corresponding to each cluster. **d** The numbers of viruses from different clusters emerged in the three groups during the outbreak. The pie charts at the right show the proportion of the clusters in the three groups. Source data are provided as a Source Data file.

and T28144C located in ORF8 was non-synonymous (L251S). After removing the clustered outbreak cases in C1 and C2 clusters, the increase in the frequency of these two mutations was still higher than that of others. Furthermore, these two SNPs co-occur in the same virus genomes in cluster C2, and dominate the other genotypes in the local transmission cohort in Beijing. We reviewed the cases and transmission chains having these two SNPs, and found 15 cases of C1 that belong to four clustered case groups, and eight cases of C2 that belong to four clustered case groups. We compared the occurrence rate of C2 in the local group with that in the Wuhan group after removing the extra cases in each group. Although the difference was not significant ($p = 0.107$), we observed a higher occurrence rate of C2 in the local group (42.9% vs. 16.7%). Whether these two SNPs are associated with virus transmissibility warrants further investigation, as this result appears to conflict with the observation that C2 and other clusters deriving from C2 are less dominant globally. As for the group of imported cases, there are four dispersive SNPs presenting with frequency >60%, and three consecutive SNPs from 28881–28883 are presenting with frequency

28%. Among the four dispersive SNPs, C241T is located in the 5′-UTR; C3037T is a synonymous mutation and C14408T is a non-synonymous mutation (P4715L) in the *ORF1ab*, and A23403G is a non-synonymous mutation (D614G) in the *S* gene. The three consecutive SNPs from 28881–28883 occur together in 11 genomes of cluster C6, and result in mutations of two consecutive amino acid residues, R203K and G204, in the N protein (Supplementary Fig. 2). Of note, we compared the clinical features and outcome of patients that harbor viruses containing the featured SNPs and did not find any statistically significant associations between the SNPs and the disease severity (Supplementary Fig. 7 and Supplementary Table 4). Overall, 248 iSNVs that present in the genome with low mutated frequencies, were discovered in our 102 samples and 240 of them are located in protein-coding regions. Among them, 162 are non-synonymous mutations and seven are stop gains (Supplementary Data 3). In general, the density of iSNVs in the genome is lower than that of SNPs (Fig. 3c), and this reflects the dynamics within host being dominated by purifying selection or a slow evolutionary rate.

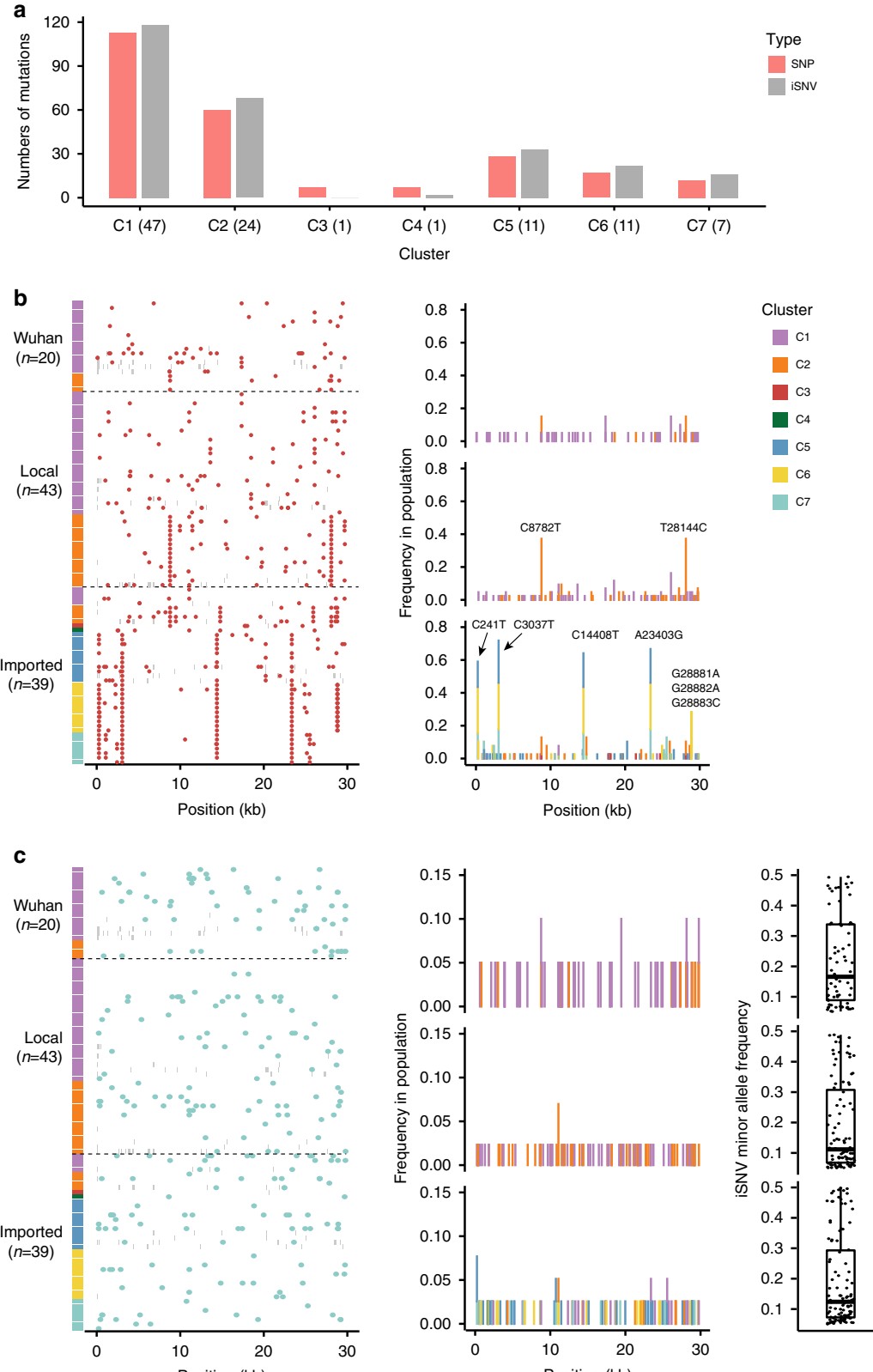

**Fig. 3 The distribution of SNPs and iSNVs in the viral genomes sequenced in this study. a** The numbers of SNPs and iSNVs in each cluster. The number of genomes in each cluster are displayed in the brackets. **b** The distribution (left) and frequency (right) of SNPs identified in the 102 genomes of the three groups and seven clusters. The red dots represent the positions of SNPs. The histogram shows the frequencies of the SNPs in each cluster. **c** The distribution (left), frequency (middle), and minor allele frequency (right) of iSNVs identified in the 102 genomes of the three groups and seven clusters. The green dots represent the positions of iSNVs. The histograms show the frequencies of the iSNVs in the enrolled population. The boxplots and scatter plots show the minor allele frequency of iSNVs. Boxplots indicate median (middle line), the first and third quartiles (box), the first quartile minus 1.5-fold the interquartile range, and the third quartile plus 1.5-fold the interquartile range. Source data are provided as a Source Data file.

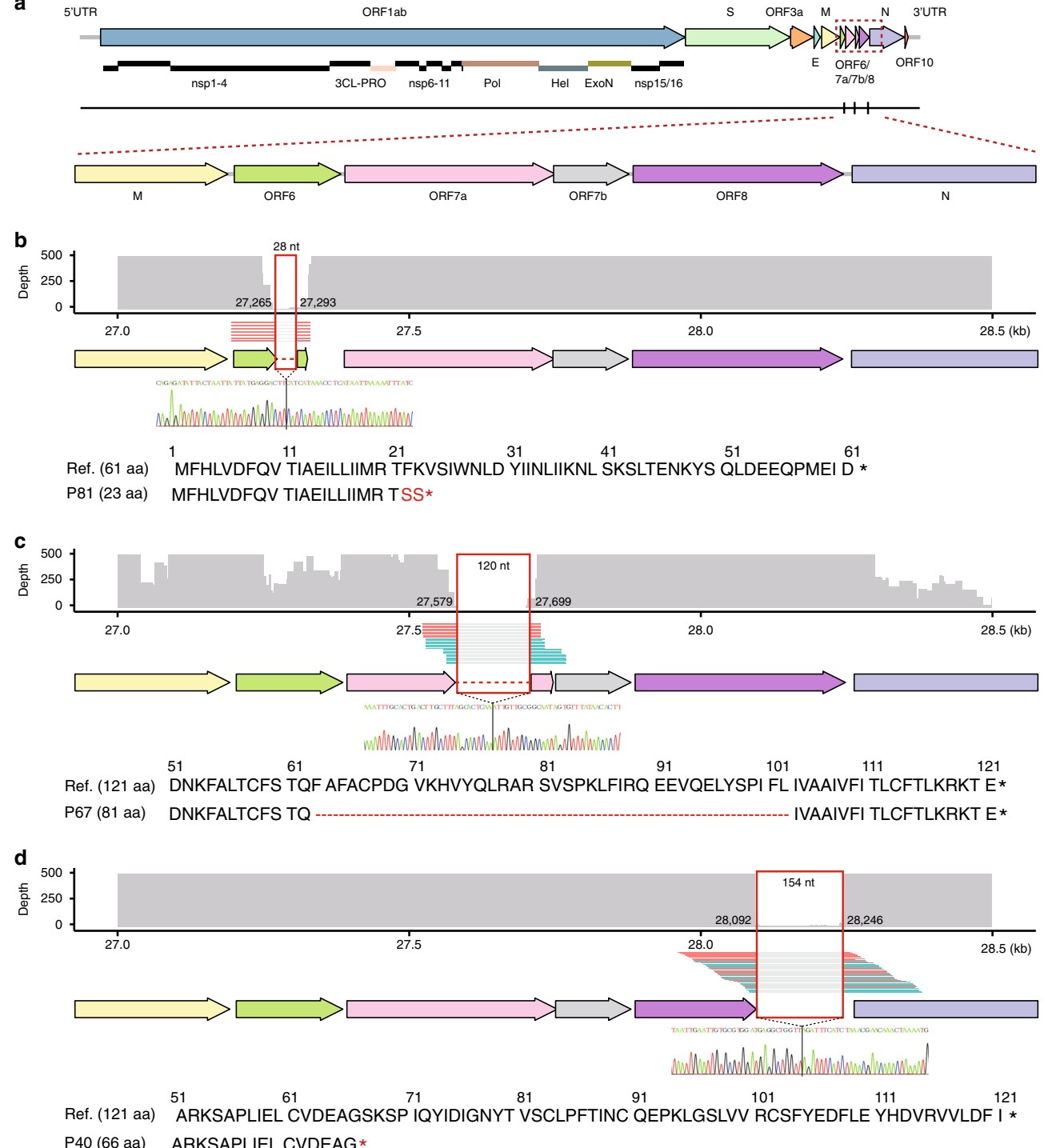

**Fig. 4 The indels identified in the viral genomes sequenced in this study. a** The loci of the indels and their distribution in the sequenced genomes. The arrow chart on top represents the protein-coding genes of SARS-CoV-2, and the vertical lines below mark the loci of the indels. **b–d** The read coverage of the three long fragment deletions in ORF6, ORF7a, and ORF8, respectively. Arrow charts in the middle represent the protein-coding genes, and the curves of read coverage and split reads supporting the deletions are displayed above. Forward and reverse mapping reads are in blue and red, respectively. Sanger sequencing results of corresponding PCR fragments are displayed below; resulting protein mutations are shown at the bottom.

**Large genetic insertion and deletion fragments in SARS-CoV-2.** Of special note, after scrutinizing the 102 high-quality genomes, we uncovered 53 possible indels (Supplementary Data 4), and three of which were long deletions (>20 bp) supported by PCR and Sanger sequencing (Fig. 4a and Supplementary Table 5).

These three long deletions were observed in three different patients. Patient P81 had a 28-nt deletion within *ORF6*, which would shorten the protein by 9-amino acids (AA) and result in early termination of translation (Fig. 4b); patient P67 had a 120-nt deletion within the *ORF7a*, which would cause a 30-AA

deletion that amounts to 33% of the whole protein (Fig. 4c); and patient P40 had a 154-nt deletion within the *ORF8*, which would lead to a premature stop codon and result in a 66-AA *ORF8* protein that is only half of its normal length (Fig. 4d). In addition, the 154-nt deletion in *ORF8* was detected in a pharyngeal swab and two fecal samples collected a week apart from patient P40, indicating that this deletion likely has a little negative effect on the survival and replication of the virus in vivo. Whether and how these indel mutants affect the virus has not been tested in the current study. Our findings based on deep sequencing could suggest high fault tolerance of the SARS-CoV-2 genome.

## Discussion

Beijing, as the capital of China and an international transportation hub, has faced threats during both disease development periods. The first period of community-linked cases started with the first recorded infection of a traveler from Wuhan on January 20, 2020, and ended on March 9, 2020, during which the disease spread by means of both Wuhan exposure and local transmission. With the control of COVID-19 in China and the outbreaks in Europe and US[16,17], overseas travelers started to arrive at the border and infected cases started to mount. The international-linked cases arrived at the Beijing border when the first two imported cases from Iran were reported on February 29, 2020. The surge of imported cases was stopped on March 23 by the redirection of inbound flights to other cities to facilitate screening. Sporadic cases from passengers in medical observation were reported since then. Until April 17, a total of 174 overseas imported cases were recorded in Beijing, and only four local cases were identified, all of which were related to imported cases.

In summary, this study describes the SARS-CoV-2 cases in the two periods in Beijing. The genomic surveillance of imported COVID-19 cases provided detailed genomic differences in a better-defined transmission map, and better evaluated the outcome of public health strategies. During the first period, chains of infections occurred and transmissions were halted by implementing strict interventions, including nucleic acid-based screening and diagnoses, self-isolation, and contact restrictions. In the second period, nearly all imported cases were caught via quarantine, screening, and detection measures. Therefore, in-time detection at borders with mandatory quarantine is an effective way to prevent recurring COVID-19 outbreak caused by importation from travelers when we tried to eliminate the infectious disease in a local geographical region[18].

Analysis of viral genomes revealed seven SNP-defined cluster types in line with the previous studies[14,15,19]. The cases in the first period in Beijing was dominated by the early types C1 and C2, while the overseas imported cases were composed of a diverse combination of cluster types. The increase of genomic diversity reflects the global transmission and following endemics[19]. Generally, random mutations were found along the SARS-CoV-2 genome, but as of yet, there is no evidence to show a correlation between these SNPs and viral adaptation. We also identified a number of interesting large indels in the viral genomes of a few specific patients. While deletions in *ORF8* had previously been found in SARS-CoV-2 and SARS-CoV by others[20,21], in this study we additionally found novel long deletions in *ORF6* and *ORF7*. Although our observation is limited by the massive undersampling of cases that have occurred globally, and more functional molecular analysis is required on these mutations, the observation also provides insights into the genome structural stability of SARS-CoV-2, and could be an important consideration in the development of gene defect vaccines[22].

## Methods

**Patients and sample collection**. This was a study of patients admitted to a Beijing hospital who tested positive for SARS-CoV-2 between January 29 and April 17, 2020. The patients were diagnosed based on the 7th guideline for the diagnosis and treatment of COVID-19 from the National Health Commission of the People's Republic of China. The pharyngeal swabs, sputum, and blood samples were collected according to the clinical guidelines. Viral RNA was extracted using the QIAamp® Viral RNA Mini Kit according to the manufacturer's instructions, except that carrier RNA was omitted to facilitate downstream high-throughput sequencing analysis. RT-PCR was conducted as previously described[23,24]. To understand the genetic characteristics of SARS-CoV-2 in Beijing, we collected pharyngeal swabs and sputa samples from the patients to perform viral genome sequencing after admission and consent procedures. In total, we collected 42 pharyngeal swabs, 63 sputa, and 21 fecal samples from 126 cases admitted to our hospital. Among these samples, the Ct values in RT-PCR of 63 samples were available and ranged from 12 to 37.52.

The following populations were considered to be of high risks during the COVID-19 outbreak in China: (i) travelers that returned from or passed by Hubei province or other regions with high epidemic activity, including all travelers from foreign countries and regions with high epidemic activity since March 2; (ii) patients with respiratory symptoms including fever, cough, sore throat, etc; (iii) close contacts with confirmed COVID-19 patients, and those exposed to virus-contaminated animals, items, or environments while not having effective protection. Epidemiological information was collected through brief interviews with each patient. Several investigators independently interviewed each patient to collect the accurate exposure histories of Hubei and overseas travel within one month.

**Sequencing of the viral genome**. We performed meta-transcriptomic sequencing to obtain the genome sequences of SARS-CoV-2 after human rRNA removal and DNA digestion. Viral RNA was extracted using the QIAamp® Viral RNA Mini Kit (Qiagen, Valencia, CA, USA) according to the manufacturer's instructions, other than the carrier RNA was omitted to facilitate downstream high-throughput sequencing analysis. We then performed rRNA removal using the MGIEasy rRNA Depletion Kit (BGI, Shenzhen, China). We used the MetagenomIc RNA EnRichment VirAl sequencing (MINERVA) approach[11,12]. This approach uses direct tagmentation of RNA/DNA hybrids using Tn5 transposase to greatly simplify the sequencing library construction process, allowed us to conduct rapid library preparation using low volume input RNA templates (5.4 µl) within 4 h. The meta-transcriptome libraries further underwent the enrichment process using biotinylated RNA probes targeting the whole viral genome (iGeneTech, Beijing, China). The final viral-enriched libraries were sequenced on an Illumina NextSeq500 in 2x75bp pair-end mode. By mapping the clean read data to the reference genome, we obtained 23.49 Mb (QRI: 9.4–54.4 Mb) high-quality viral reads per sample. By the quality control with sequencing depth ≥ 5 and reference coverage ≥ 85%, we obtained enough data to generate high-quality viral genomes in 102 samples. Finally, we used these high-quality data to perform genomic analysis along with published data.

PCR, sequencing errors, and the potential contaminants induced in metagenomic sequencing always the big concern of next-generation sequencing. To better detect these contaminants and sequencing error, we enrolled negative control of water in each run and several pharyngeal swabs from the healthy in random runs to confirm whether we have false-positive of the virus in the study.

**Genomic characteristics of published data enrolled**. We downloaded the published SARS-CoV-2 genomes from GISAID (Global Initiative on Sharing All Influenza Data, https://www.gisaid.org/)[13]. We would like to thank all the authors who have kindly deposited and shared genome data on GISAID. A table with genome sequence acknowledgments can be found in Supplementary Data 2. A total of 7205 SARS-CoV-2 sequences on GISAID on 12th Apr 2020 were downloaded. According to the National Genomics Data Center (https://bigd.big.ac.cn/), high-quality sequences were (1) completely assembled, (2) anonymous nucleotides ≤15, (3) degenerate bases ≤50, (4) gaps (including deletion and insertion) ≤2, (5) coverage of mutation regions < 25%, (6) sequence length ≥25,000 bp. The resulting data set of 4118 sequences were finally included to represent the global diversity of the virus from six continents, including Africa (31 genomes), Asia (511), Europe (1976), North America (1229), South America (23), and Oceania (348).

**Calling SNP and iSNV**. Quality control, error correction, calling of iSNVs, and SNPs from the sequencing read, we obtained was performed as following[25]. Firstly, only high-quality samples and high-quality sites were included for the consideration of SNPs and iSNVs. The sites and the samples were determined as the following steps: (1) sequencing reads were pair-ended aligned to the reference genome sequence (GenBank accession no. NC_045512.2) using Bowtie2 v2.1.0 by default parameters and the alignments were reformatted using SAMtools v1.3.1;[26,27] (2) for each site of the SARS-CoV-2 genome, the aligned low-quality bases (<Q20) and indels were excluded to reduce possible false positives and the site depth and strand bias was re-calculated; (3) samples with more than 3,000 sites with a

sequencing depth ≥100× were selected as candidate samples. The consensus sequences were generated from the read mapping results by bowtie2 and SAMtools using the consensus method in BCFtools v0.1.19 according to the reference.

Calling iSNVs by our sequencing data should meet the following criteria to ensure high quality: (1) depth of the minor allele of ≥5; (2) strand bias of the minor allele less than tenfold; (3) Positions 1–55 and 29,804–29,903 were removed; and (4) the sites that appear to be highly homoplastic were masked (https://virological.org/t/issues-with-sars-cov-2-sequencing-data/473), except the common SNPs present in >5% of the genomes enrolled in our study. Here, we defined iSNVs with the minor allele frequency of ≥5% (a conservative cutoff based on an error rate estimation). Single nucleotide polymorphisms (SNPs) were defined with the minor allele frequency of <5%, the mutation allele frequency should be >95%, and sequencing depth ≥5. SNPs in public genomes were detected by the comparison to the reference genome sequence (GenBank accession no. NC_045512.2) using BLAST v2.2.18, MView v1.67, and inhouse perl script[28,29].

**Phylogenetic analysis and lineage determination**. The consensus sequences we obtained and the published genomes from GISAID were aligned with the reference genome using MAFFT v7.453[30]. We trimmed the genome positions covered by <90% of the sequences in the alignment to prone low-coverage positions. The General Time Reversible (GTR) + F + R2 model was selected to build the maximum likelihood phylogenetic tree based on Bayesian information criteria (BIC) score generated by IQ-TREE v1.6.12[31], which was further used for phylogenetic tree construction. Lineage determination was performed by pangolin v2.0.4 using the default parameters (https://github.com/hCoV-2019/pangolin)[14].

**Clustering analysis based on SNPs**. The SNPs from our data and those from published genomes were combined together firstly. The high-frequency SNPs (hfSNPs) were defined as common SNPs presented in >5% genomes. We performed clustering analysis to distinguish the genomes using the following methods in R "stats" package. The distance matrix was defined with "dist" using the Euclidean distances among the genomes, which were calculated by the presence of hfSNPs. Hierarchical cluster analysis and tree view were used for better review of the relationship among the clusters, where we selected the tree height with 20 by "cutree".

Clustered infection was defined as a concentration of infections in the same family or company. The differential distribution of C1 and C2 between Wuhan and local groups was tested with two-sided Fisher's exact test using the "stats" package in R. The patients from the same clustered infection were considered as one case.

**Ethics declarations**. This study was approved by the Ethics Committee of Beijing Ditan Hospital, Capital Medical University (KT2020-006-01). The study conformed to the ethical guidelines of the 1975 Declaration of Helsinki.

**Reporting summary**. Further information on research design is available in the Nature Research Reporting Summary linked to this article.

## Data availability

The viral sequencing data we obtained in this study have been deposited in the GenBank database under the accession number PRJNA667180. The published SARS-CoV-2 genomes used in this study were downloaded from GISAID (https://www.gisaid.org/) with suitable acknowledgments (Supplementary Data 2). Source data are provided with this paper.

## Code availability

The source code for SNP and iSNV calling can be accessed at GitHub (http://github.com/generality/iSNV-calling/).

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

## Acknowledgements

We thank all health care workers involved in the diagnosis and treatment of the COVID-19 patients in Beijing Ditan Hospital, Capital Medical University.

## Author contributions

P.D., N.D., J.L., J.C., and H.L. performed the sequence data analysis. Q.W., Z.C., W.X., J.L., G.G., and S.W. collected and analyzed the clinic data. C.S., K.H., M.H., F.Y., Lin Wang, W.W., K.A., Y.H., and J.W. performed most of the experiments. A.W., D.L., L.Wang, L.Wei, and S.M. provided intellectual input and helped to interpret the data. P.D., C.C., D.L., J.W., F.Z., and H.Z. wrote the manuscript. All of the authors discussed the results and commented on the manuscript. C.C., Z.H., F.Z., D.L., and J.W. supervised the study.

## Competing interests

The authors declare no competing interests.
