## [Peer Review File · Nature Communications]

REVIEWER COMMENTS

Reviewer #1 (Remarks to the Author):

The study by Du and colleagues examines the SARS-CoV-2 epidemic in Beijing, China was it experienced two separate sources of transmission, one largely driven by domestic introductions from Wuhan, and a subsequent spike in cases as the result of international introductions from Europe and North America. The authors were able to generate complete or nearly complete genome sequences from >30% of the cases diagnosed in Beijing, representing a robust genomic dataset. Authors were able to use high frequency SNPs to classify each genome into separate clusters to track temporal diversity of SARS-CoV-2 in Beijing. The authors also describe various aspects of these samples including large, verified deletions. Collectively, this represents a thorough analysis of the genomic diversity of SARS-CoV-2 in a major city, and shows how swift control measures can stop ongoing transmission. However, there are aspects of the manuscript that need to be clarified prior to publication in Nature Communications. Below are both major and minor points.

Major Points

-While I do not disagree with the cluster analysis, it does not seem to add anything to the paper that a simple maximum likelihood phylogenetic approach would not. For example, Cluster3 corresponds to lineage A.1 (using the Pangolin system (tool here: <https://github.com/cov-lineages/pangolin> paper here: <https://www.biorxiv.org/content/10.1101/2020.04.17.046086v1>) Cluster2 corresponds to lineage A, The split from C1 to C5 represents the split between A and B lineages, etc. I believe that adding yet another way of classifying different viral genotypes is unnecessarily convoluted and not terribly beneficial. The global genetic diversity of SARS-CoV-2 is well-defined and describing these virus genotypes as belonging to a lineage rather than a cluster would go a long way in improving the understanding of the manuscript. This is especially pertinent when the authors discuss the global distribution of these viral genotypes (e.g. on line 252-254). I don't think this is absolutely necessary for the publication of the manuscript, but would help clarify the genomes described in the paper.

-I am confused by the discussion of SNPs and iSNVs. On line 173-175, the authors state they revealed 3,309 SNPs compared to the reference Wuhan-Hu-1 genome. I think it is important to distinguish between a consensus sequence SNP and a single nucleotide variant. In the methods the authors state (Line 372-374) "The single nucleotide polymorphisms (SNPs) were identified using the similar approach and criteria, except the mutation allele frequency should be >95% for SNPs." By this criteria, there cannot be over 3,000 SNPs compared to the reference genome in all of the global SARS-CoV-2 sequences. I think the authors are refereeing to sites that show some level of intra-host variability, which is commonly referred to as a SNV. This point needs to be clarified. As well, on Line 229 the authors state "In total, we identified 227 SNPs, which is roughly 3-7 SNPs per patient; of these, 76 are synonymous and 138 are non-synonymous substitutions" Similarly, the authors also talk about iSNVs on Line 262-269. Im not sure where the differences are here. The authors should clarify what criteria they are using to classify SNPs and iSNVs or hfSNPs, and this needs to be consistent throughout the manuscript.

-Sequencing controls need to be discussed, especially if discussing intra-sample variants. What controls were used to ensure there was no cross contamination on the sequencing runs?

Minor points

-Line 86- "based" Line 98- "have" Line 112 "Initiated" Line 132 "Overseas" Line 134 "Travelers"

-How was the clustering analysis performed? I think a more thorough discussion other than the R package used is warranted.

-The authors attempt to compare their classification to others being used on lines 394-403. However, the "A, B, and C types" described by Foster et al. are not a relevant classification scheme as the analysis was highly flawed and only encompasses a fraction of the number of sequences currently available. Again, I would highly suggest adapting the naming of clusters in this manuscript to lineages through a phylogenetic analysis to improve clarity.

-Line 188- saying a genome was "derived from" in these clusters implies that a transmission chain resulted in these new genotypes (i.e. mutations occurred as a result of sustained transmission in Beijing). Opposed to these genotypes having a global spread and being imported. Again, this would be clarified using a common lineage naming scheme.

-The results sections describe an analysis to test for selection in these genomes, but there is no discussion in the methods on how this was performed.

Reviewer #2 (Remarks to the Author):

Du et al. describes the genomic analysis of 102 COVID-19 cases in Beijing, China at one of the hospitals receiving and treating COVID-19 patients. Various observations are presented but most findings remain speculative.

Major comments:

The authors comment on two periods of COVID-19 outbreak but I am not sure if these periods are widely recognised as such. Please clarify if this delimitation is done by the authors for the sake of their own narrative or if it done somewhere else. There is a citation for this, but it is clearly a mistake (reference number 6 directs us to a paper on Clostridium from 2005).

It is not clear to me why the 'second wave' is delineated as such. The authors underscore that this 'wave' started with the first report of an overseas introduction but there was not a substantial surge in community-linked cases—4 in total. Actually, it sounds like the control measures are delaying another 'wave' despite an important number of returning infected travellers (as figs A-B point out too). My recommendation is to refrain from calling it a 'wave' and describe such surveillance findings as international-linked cases (as the title says).

The number of samples doesn't match: 55 Wuhan travellers + 81 linked to Wuhan travellers + 1 linked to the latter + 121 international travellers + 4 linked to international travellers = 262 but 273 cases reported by the authors; 187 samples processed and 102 passes QC—which ones.

L174 3,309 SNPs is a massive number of SNPs for Beijing only. Could the authors verify this. For reference, the study of Miller et al in Israel reported 222 using 212 sequences. L229 mentions 227 SNPs, please clarify.

L175-L205 attempts to classify the global diversity into clusters. However, widely disseminated nomenclatures exist out there; moreover, the authors comment that C1 and C2 were previously named L-type and S-type, respectively. I would refrain to propose new classifications without a solid rationale for doing so (in order to make naming of isolates more tractable across papers) and instead I would use the clusters naming from the literature (for example, GISAID has a nomenclature based on presence/absence of mutations and Rambaut et al. has a nomenclature guided by phylogenetic relationships—hence shared mutations by ancestry)

L207-L224 these observations are expected and so the text could be shortened as it does not add much—whereas in Wuhan depletion of susceptibles and lockdowns in place limit the extent of virus

diversification, the global spread of the virus provided enough room for mutations to arise and accumulate as it spread.

L233-261. This is too speculative and the author did not provide details on the curation of the data (also, L170 please specify how representative samples from GISAD were selected).

It is not clear from the text if some problematic positions from the alignments were masked (e.g. homoplasic positions that are exclusive to a single sequencing lab or geographic location and might be caused by a common source of error—<https://virological.org/t/issues-with-sars-cov-2-sequencing-data/473>). Data curation is highly relevant for this kind of studies as the error rate of some genomic generation techniques is close to that of the virus evolution. Moreover, please note that the global genome availability is heterogenous and widespread lineages are probably the outcome of founder effects. For instance, it sounds like observations in L240 reflects the very limited diversity of SARS-CoV-2 at the time and not differences in transmissibility. Also, the international-linked cases did not result in widespread community transmission, so the authors are actually analysing a sub-sample of the global genetic pool—at present, and based on more thorough analysis there is no evidence of distinct strains.

L268 or a slow evolutionary rate?

Finally, the authors should underscore how the genomic surveillance of imported COVID-19 cases in Beijing contribute to outbreak management.

Minor comments

L88 sentence reads incomplete

Please check if 'immigration' [control] is intended rather than 'customs' [control]

L111 "went through both periods of COVID-19 outbreak, and thus experienced"?

L130 reported cases

L152 returning Chinese travelers.

L153 maybe slightly better 'reallocation' or 'redistribution'. Please also classify if this was done to facilitate screening.

Reviewer #3 (Remarks to the Author):

Du et al. describe the genomic epidemiology of COVID-19 cases that were imported into Beijing. Their results nicely demonstrate two waves of infection, both the domestic wave from Wuhan and later the global wave from international travellers. The figures are clear and provide a very good explanation of the results.

While this manuscript shows interesting results, it needs some improvement. Notably, the writing requires revision to improve grammar and general sentence structure.

The authors state that genomic sequencing was performed on 31.5% of the cases in Beijing. First, the dates between which this is true should be stated since this is likely to drastically change. Second, from what I understand, only 100 genomes were actually sequenced, not 187 as the abstract notes. Please clarify.

The authors have annotated clusters as C1-C7. It might be worthwhile to also annotate these with the suggested lineage names by Rambaut et al. 2020 so that this diversity can be more easily compared to diversity seen in other regions.

Could the authors explain why they use a metagenomic sequencing approach here as opposed to genome sequencing, what benefits it has, etc.?

Response to comments

Reviewer #1:

Major Points

1. Q: While I do not disagree with the cluster analysis, it does not seem to add anything to the paper that a simple maximum likelihood phylogenetic approach would not. For example, Cluster3 corresponds to lineage A.1 (using the Pangolin system (tool here: <https://github.com/cov-lineages/pangolin> paper here: <https://www.biorxiv.org/content/10.1101/2020.04.17.046086v1>) Cluster2 corresponds to lineage A, The split from C1 to C5 represents the split between A and B lineages, etc. I believe that adding yet another way of classifying different viral genotypes is unnecessarily convoluted and not terribly beneficial. The global genetic diversity of SARS-CoV-2 is well-defined and describing these virus genotypes as belonging to a lineage rather than a cluster would go a long way in improving the understanding of the manuscript. This is especially pertinent when the authors discuss the global distribution of these viral genotypes (e.g. on line 252-254). I don't think this is absolutely necessary for the publication of the manuscript, but would help clarify the genomes described in the paper.

A: We thank the reviewer for the valuable comments. We agree that it is necessary to perform phylogenetic analysis with the published SARS-CoV-2 genomes and provide more scientifically accurate results. We have performed both phylogenetic analysis and clustering analysis in the genomes, and compared the results with the results of Forster P et al (PNAS, 2020) in the previous version (the paper of Rambaut et al had not been accepted when we performed the analysis). According to the suggestions and Rambaut's method, we have provided a new lineage nomenclature in the revised manuscript (Fig. 2, Fig. S3, Fig. S4 and Line 177-181).

The main aim of this manuscript is to better illustrate the viral diversity of SARS-CoV-2 in Beijing. The clustering analysis could better exhibit the difference of groups, and the following minimal spanning tree displayed the relationship among the groups. Considering that 1) the sequences of our samples scattered widely in the

phylogenetic tree and the sizes of different branches greatly varied, 2) about 55 out of 102 (~53.9%) sequences could not be denoted at the sublevels based on the Rambaut's lineage nomenclature (Fig. S4), we displayed both results of clustering analysis (Fig. 2, Fig. S4) and phylogenetic analysis (Fig. S3) in this new version.

2. Q: I am confused by the discussion of SNPs and iSNVs. On line 173-175, the authors state they revealed 3,309 SNPs compared to the reference Wuhan-Hu-1 genome. I think it is important to distinguish between a consensus sequence SNP and a single nucleotide variant. In the methods the authors state (Line 372-374) "The single nucleotide polymorphisms (SNPs) were identified using the similar approach and criteria, except the mutation allele frequency should be >95% for SNPs." By this criteria, there cannot be over 3,000 SNPs compared to the reference genome in all of the global SARS-CoV-2 sequences. I think the authors are refereeing to sites that show some level of intra-host variability, which is commonly referred to as a SNV. This point needs to be clarified. As well, on Line 229 the authors state "In total, we identified 227 SNPs, which is roughly 3-7 SNPs per patient; of these, 76 are synonymous and 138 are non-synonymous substitutions" Similarly, the authors also talk about iSNVs on Line 262-269. I'm not sure where the differences are here. The authors should clarify what criteria they are using to classify SNPs and iSNVs or hfSNPs, and this needs to be consistent throughout the manuscript.

A: We apologize for these confusions. We obtained the 4118 global genomes as consensus sequences, and detected 2,752 variations occurred in 2,669 SNP sites from these data. Since these sequences were not curated, there could be more variations than expected. We also detected 492 variations in 221 SNP sites that had >95% mutant allele frequencies in our 102 samples. After removing duplicates, we obtained 2,795 SNPs in total. We also detected iSNV with 5-95% mutant allele frequencies in our 102 samples. We have clarified the criteria we used to classify SNPs and iSNVs, and the definition of hfSNPs in Methods on Line 382-392 and 404-407.

Calling iSNVs by our sequencing data should meet the following criteria to ensure high quality: (1) depth of the minor allele of ≥ 5 ; (2) strand bias of the minor allele

less than tenfold; (3) Positions 1–55 and 29804–29903 were removed; and (4) the sites that appear to be highly homoplastic were masked, except the common SNPs present in >5% of the genomes enrolled in our study. Here, we defined iSNVs with the minor allele frequency of $\geq 5\%$ (a conservative cutoff based on an error rate estimation). Single nucleotide polymorphisms (SNPs) were defined with the minor allele frequency of <5%, the mutation allele frequency should be >95%, and sequencing depth ≥ 5 . SNPs in public genomes were detected by the comparison to the reference genome sequence (GenBank accession no. NC_045512.2) using BLAST, MView and inhouse perl script.

The SNPs from our data and those from published genomes were combined together firstly. The high-frequency SNPs (hfSNPs) were defined as common SNPs presented in >5% genomes.

All information has been updated in the method, and thanks for the suggestions.

3. Q: Sequencing controls need to be discussed, especially if discussing intra-sample variants. What controls were used to ensure there was no cross contamination on the sequencing runs?

We thank the reviewer for this comment. The negative control is enrolled in our study. We have added the description of controls in method about what we used in this study on Line 348-352 to clarify control sample setting.

PCR, sequencing errors and the potential contaminants induced in metagenomic sequencing always the big concern of next generation sequencing. To better detect these contaminants and sequencing error, we enrolled negative control of water in each run and several pharyngeal swabs from the healthy in random runs to confirm whether we have false-positive of virus in the study.

Minor points

4. Q: Line 86- “based” Line 98- “have” Line 112 “Initiated” Line 132 “Overseas”
Line 134 “Travelers”

A: We apologize for these errors and have revised the whole manuscript.

5. Q: How was the clustering analysis performed? I think a more thorough discussion other than the R package used is warranted.

A: We have added more description in the Methods part on Line 405-410.

We performed clustering analysis to distinguish the genomes using the following methods and functions from the stats package in R. The Euclidean distances among the genomes were calculated by hierarchical cluster analysis firstly based on the presence of hfSNPs using dist function, then the genomes were clustered based on the Euclidean distances using the ward.D2 method. Finally, the clusters were identified based on the clustering results using cutree function (h=20).

6. Q: The authors attempt to compare their classification to others being used on lines 394-403. However, the “A, B, and C types” described by Foster et al. are not a relevant classification scheme as the analysis was highly flawed and only encompasses a fraction of the number of sequences currently available. Again, I would highly suggest adapting the naming of clusters in this manuscript to lineages through a phylogenetic analysis to improve clarity.

A: We thank the reviewer for pointing out this issue. We have removed the corresponding part and added the phylogenetic analysis.

7. Q: Line 188- saying a genome was “derived from” in these clusters implies that a transmission chain resulted in these new genotypes (i.e. mutations occurred as a result of sustained transmission in Beijing). Opposed to these genotypes having a global spread and being imported. Again, this would be clarified using a common lineage naming scheme.

A: We thank the reviewer for this suggestion and have revised the manuscript.

8. Q: The results sections describe an analysis to test for selection in these genomes, but there is no discussion in the methods on how this was performed.

A: We apologize for that and we have added the methods in Methods on Line 413-416.

Clustered infection was defined as a concentration of infections in a same family or company. The differential distribution of C1 and C2 between Wuhan and local groups was tested with Fisher's exact test using the stats package in R. The patients from the same clustered infection were considered as one case.

Reviewer #2:

1. Q: Major comments:

The authors comment on two periods of COVID-19 outbreak but I am not sure if these periods are widely recognized as such. Please clarify if this delimitation is done by the authors for the sake of their own narrative or if it done somewhere else. There is a citation for this, but it is clearly a mistake (reference number 6 directs us to a paper on Clostridium from 2005).

A: We apologize for the confusion. The two periods of COVID-19 outbreak came from the Figure 1 of WHO situation report 71 on March 31, 2020. From the report the regional outbreak occurred mainly in Western Pacific in February 2020, then the pandemic all over the world (declared by WHO on March 11, 2020). Besides, we define the two periods to describe the outbreak dynamics in Beijing. We have added the reference of WHO situation report and revised the manuscript for clarification.

(https://www.who.int/docs/default-source/coronaviruse/situation-reports/20200331-sit-rep-71-covid-19.pdf?sfvrsn=4360e92b_8)

2. Q: It is not clear to me why the ‘second wave’ is delineated as such. The authors underscore that this ‘wave’ started with the first report of an overseas introduction but there was not a substantial surge in community-linked cases—4 in total. Actually, it sounds like the control measures are delaying another ‘wave’ despite an important number of returning infected travelers (as figs A-B point out too). My recommendation is to refrain from call it a ‘wave’ and describe such surveillance findings as international-linked cases (as the title says).

A: We appreciate the reviewer’s valuable comments. Indeed, the number of community-linked cases was suppressed to a minimal level by tight border control. By ‘the second wave’, we meant to describe the international-linked cases, which could cause confusion. According to the suggestion, we have revised the manuscript to clarify the two periods.

3. Q: The number of samples doesn’t match: 55 wuhan travelers + 81 linked to wuhan travelers + 1 linked to the latter + 121 international travelers + 4 linked to international travelers=262 but 273 cases reported by the authors; 187 samples processed and 102 passes QC—which ones.

A: We apologize for wrong expression. The total number of patients in our hospital was 273: 55 Wuhan travelers + 81 linked to Wuhan travelers + 1 linked to the latter + 11 without clear epidemiological links with other cases + 121 international travelers + 4 linked to international travelers. We collected 187 samples from some of patients and obtained 102 high quality datasets. We have revised the manuscript.

4. Q: L174 3,309 SNPs is a massive number of SNPs for Beijing only. Could the authors verify this. For reference, the study of Miller et al in Israel reported 222 using used 212 sequences. L229 mentions 227 SNPs, please clarify.

A: We apologize for these confusions. We obtained the 4118 global genomes as consensus sequences, and detected 2,752 variations occurred in 2,669 SNP sites from these data. Since these sequences were not curated, there could be more variations than expected. We also detected 492 variations in 221 SNP sites that had >95% mutant allele frequencies in our 102 samples. After removing duplicates, we total obtained 2,795 (2,669 + 221 and then removing repeated sites) SNP sites in genome scale.

5. Q: L175-L205 attempts to classify the global diversity into clusters. However widely disseminated nomenclatures exist out there; moreover, the authors comment that C1 and C2 were previously named L-type and S-type, respectively. I would refrain to propose new classifications without a solid rationale for doing so (in order to make naming of isolates more tractable across papers) and instead I would use the clusters naming from the literature (for example, GISAID has a nomenclature based on presence/absence of mutations and Rambaut et al. has a nomenclature guided by phylogenetic relationships—hence shared mutations by ancestry)

A: We thank the reviewer for this valuable suggestion. Indeed, performing phylogenetic analysis with the published SARS-CoV-2 genomes is more scientifically accurate and this is exactly what we did. The main reason why we switched to the clustering strategy was because our sequences scattered widely in the phylogenetic tree, and the sizes of different branches greatly varied. In addition, about 55 out 102 (~53.9%) sequences could not be denoted at the sublevels based on the Rambaut's lineage nomenclature. Therefore, we have put the phylogenetic analysis result back to the manuscript (Fig. 2A, 2C, S3 and S4) and also added the comparison between phylogenetic tree and clusters (Line 177-181).

6. Q: L207-L224 these observations are expected and so the text could be shortened as it does not add much—whereas in Wuhan depletion of susceptibles and lockdowns in place limit the extent of virus diversification, the global spread of the virus provided enough room for mutations to arise and accumulate as it spread.

A: We thank the reviewer for this suggestion and have revised the manuscript.

7. Q: L233-261. This is too speculative and the author did not provide details on the curation of the data (also, L170 please specify how representative samples from GISAD were selected).

It is not clear from the text if some problematic positions from the alignments were masked (e.g. homoplasic positions that are exclusive to a single sequencing lab or geographic location and might be caused by a common source of error—<https://virological.org/t/issues-with-sars-cov-2-sequencing-data/473>). Data curation is highly relevant for this kind of studies as the error rate of some genomic generation techniques is close to that of the virus evolution. Moreover, please note that the global genome availability is heterogenous and widespread lineages are probably the outcome of founder effects. For instance, it sounds like observations in L240 reflects the very limited diversity of SARS-CoV-2 at the time and not differences in transmissibility. Also, the international-linked cases did not result in widespread community transmission, so the authors are actually analyzing a sub-sample of the global genetic pool—at present, and based on more thorough analysis there is no evidence of distinct strains.

A: We thank the reviewer for the comments. We have added more details of bioinformatic process in Method. According to the webpage discussing the sequencing errors, three categories of mutations were suggested to remove from the phylogenetic analysis, including: (1) sites that appear to be highly homoplasic and have no phylogenetic signal and/or low prevalence, (2) homoplasic positions that are exclusive to a single sequencing lab or geographic location, (3) positions that, despite having strong phylogenetic signal, are also strongly homoplasic. We have removed two mutations in category (2), and 23 low occurrence mutations in category (1) and (3). We also masked a mutation in positions 1–55 and 29,804–29,903. However, we retained five highly occurrence sites, which were also retained in phylogenetic analysis by Rambaut et al (Nature microbiology, 2020). We have marked these sites and described in Methods.

We agree that our data were only sub-sample of the global genetic pool. The phylogenetic analysis showed more distinct lineage. Considering the major aim of this study is to provide a clear epidemiological genomic data in Beijing but not to distinct strains. we have simplified the lineages representation of our data by clusters.

8. Q: L268 or a slow evolutionary rate?

A: We thank the reviewer for this kind suggestion. We have revised this sentence accordingly on Line 252-254.

9. Q: Finally, the authors should underscore how the genomic surveillance of imported COVID-19 cases in Beijing contribute to outbreak management.

A: We thank the reviewer for this kind suggestion. We have revised the manuscript to include the contribution of infection surveillance of imported COVID-19 cases in Beijing to outbreak management emphatically on Line 274-276.

Minor comments

L88 sentence reads incomplete

Please check is 'immigration' [control] is intended rather than 'customs' [control]

L111 "went through both periods of COVID-19 outbreak, and thus experienced"?

L130 reported cases

L152 returning Chinese travelers.

L153 maybe slightly better 'reallocation' or 'redistribution'. Please also classify if this was done to facilitate screening.

A: We apologize for these errors and have revised the whole manuscript.

Reviewer #3:

1. Q: While this manuscript shows interesting results, it needs some improvement.

Notably, the writing requires revision to improve grammar and general sentence

structure.

A: We apologize for these errors and have revised the whole manuscript.

2. Q: The authors state that genomic sequencing was performed on 31.5% of the cases in Beijing. First, the dates between which this is true should be stated since this is likely to drastically change. Second, from what I understand, only 100 genomes were actually sequenced, not 187 as the abstract notes. Please clarify.

A: We are sorry for these errors. In total, 273 cases were admitted to our hospital from January 29 to April 17, 2020. We collected samples from 187 cases and obtained high quality genomes from 102 cases. We have clarified these numbers in abstract and the main text.

3. Q: The authors have annotated clusters as C1-C7. It might be worthwhile to also annotate these with the suggested lineage names by Rambaut et al. 2020 so that this diversity can be more easily compared to diversity seen in other regions.

A: We thank the reviewer for this valuable suggestion. Indeed, performing phylogenetic analysis with the published SARS-CoV-2 genomes is more scientifically accurate and this is exactly what we did. The main reason why we switched to the clustering strategy was because our sequences scattered widely in the phylogenetic tree, and the sizes of different branches greatly varied. In addition, about 55 out of 102 (~53.9%) sequences could not be denoted at the sublevels based on the Rambaut's lineage nomenclature. Therefore, we have put the phylogenetic analysis result back to the manuscript (Fig. 2A, 2C, S3 and S4) and also added the comparison between phylogenetic tree and clusters (Line 177-181).

4. Q: Could the authors explain why they use a metagenomic sequencing approach here as opposed to genome sequencing, what benefits it has, etc.?

A: We are sorry for the confusion. We employed a versatile sequencing library construction pipeline to study all COVID-19 samples. The pipeline starts with a metagenomic strategy to include as much genetic information as possible. We then

performed targeted SARS-CoV-2 sequence enrichment for deep sequencing. Though metagenomic analysis is not the emphasis of this study, we describe the full pipeline for scientific accuracy.

REVIEWERS' COMMENTS

Reviewer #1 (Remarks to the Author):

The authors have addressed all of my major concerns about the manuscript. In particular, I appreciate the authors adjusting their nomenclature to fit what is more broadly used to describe viral lineages in the literature. I find it really useful to look at Fig 2A and S3 to try and see how the viral lineages identified in the study relate to other lineages around the world. I also appreciate the clarification on SNPS, iSNVs, and the hfSNPS used for the clustering analysis. I have a few additional recommendations that I hope are easily addressable.

-Line 64 (and other portions of the manuscript where this is explicitly mentioned)- genomic sequencing was performed on 17.2% of the total identified cases in Beijing. This is a crucial point because not every case that actually occurred was identified through surveillance. Please adjust to say "total identified cases in Beijing".

-For the epidemiological sorting of cases, local, Wuhan, or imported, were these identified as such using contact tracing efforts?

-Line 184-195 and Figure S6- it is not appropriate to conclude that the clusters of cases that were identified in Beijing (C1-C7) were distributed around the world. The inverse of this statement is actually true. Line 184-195 and Figure S6 suggest that these clusters were distributed around the world, opposed to virus lineages from around the world made up these clusters. The clustering analysis was done to exhibit how cases in Beijing are grouping together and are related, and is appropriate for this purpose. Saying that these clusters are distributed throughout the world suggests to me that they were subsequently exported out of Beijing and are now being readily transmitted, which is not the case. I would remove this language or refer to the pangolin lineages that make up these clusters when you are discussing their global distribution.

-Line 235-237 discussing the admissibility of any variant is precarious without empirical laboratory data, and the authors appropriately state this observation warrants further discussion. I'd like to see the authors make note here that this observation is limited by the massive undersampling of cases that have occurred globally and it is proportional to the included genomes from GISAID in their comparisons.

Response to comments

Reviewer #1:

The authors have addressed all of my major concerns about the manuscript. In particular, I appreciate the authors adjusting their nomenclature to fit what is more broadly used to describe viral lineages in the literature. I find it really useful to look at Fig 2A and S3 to try and see how the viral lineages identified in the study relate to other lineages around the world. I also appreciate the clarification on SNPS, iSNVs, and the hfSNPS used for the clustering analysis. I have a few additional recommendations that I hope are easily addressable.

-Line 64 (and other portions of the manuscript where this is explicitly mentioned)- genomic sequencing was performed on 17.2% of the total identified cases in Beijing. This is a crucial point because not every case that actually occurred was identified through surveillance. Please adjust to say "total identified cases in Beijing".

A: We thank the reviewer for this suggestion and have revised this issue by removing the word "the".

-For the epidemiological sorting of cases, local, Wuhan, or imported, were these identified as such using contact tracing efforts?

A: The epidemiological sorting of cases was performed by interviewing the patients, and we have added that in Method section on Line 333-336.

-Line 184-195 and Figure S6- it is not appropriate to conclude that the clusters of cases that were identified in Beijing (C1-C7) were distributed around the world. The inverse of this statement is actually true. Line 184-195 and Figure S6 suggest that these clusters were distributed around the world, opposed to virus lineages from around the world made up these clusters. The clustering analysis was done to exhibit how cases in Beijing are grouping together and are related, and is appropriate for this purpose. Saying that these clusters are distributed throughout the world suggests to

me that they were subsequently exported out of Beijing and are now being readily transmitted, which is not the case. I would remove this language or refer to the pangolin lineages that make up these clusters when you are discussing their global distribution.

A: Thank you for the suggestion. Accordingly, we have removed the sentence and rephrased the context of this paragraph on Line 188 and 202.

-Line 235-237 discussing the admissibility of any variant is precarious without empirical laboratory data, and the authors appropriately state this observation warrants further discussion. I'd like to see the authors make note here that this observation is limited by the massive undersampling of cases that have occurred globally and it is proportional to the included genomes from GISAID in their comparisons.

A: Thank you for the suggestion. We have discussed the limitation in Discussion section on Line 303-305.